# The Identification and Analysis of the Centers of Geographical Public Opinions in Flood Disasters Based on Improved Naïve Bayes Network

**DOI:** 10.3390/ijerph191710809

**Published:** 2022-08-30

**Authors:** Heng Tang, Hanwei Xu, Xiaoping Rui, Xuebiao Heng, Ying Song

**Affiliations:** 1College of Hydrology and Water Resources, Hohai University, Nanjing 210024, China; 2School of Earth Sciences and Engineering, Hohai University, Nanjing 211100, China

**Keywords:** flood disasters, centers of geographic public opinions, improved naïve Bayes networks, ensemble learning, text classification, social big data

## Abstract

The increasing frequency of floods and the lack of protective measures have the potential to cause severe damage. Working from the perspective of network public opinion is an effective way to understand flood disasters. However, the existing research tends to focus on a single perspective, such as the characteristics of the text, algorithm optimization, or spatial location recognition, while scholars have paid much less attention to the impact of social-psychological differences in space on network public opinion. This research is based on the following hypothesis: When public opinions break out, the differences of network public opinions in geography will form spatially different centers of geographical public opinions in flood disasters (CGeoPOFDs). These centers represent the cities that receive the most attention from network public opinion. Based on this hypothesis, this study proposes a new way of identifying and analyzing CGeoPOFDs. First, two optimization strategies were applied to enhance a naïve Bayes network: syntactic parsing, which was used to optimize the selection of feature word vectors, and ensemble learning, which enabled multi-classifier fusion optimization. Social media data were classified through the improved algorithm, and then, various methods (hotspot analysis, geographic mapping, and sentiment analysis) were used to identify CGeoPOFDs. Finally, analysis was performed in terms of spatiotemporal, virtual, and real dimensions. In addition, microblog social data and real disaster data were used to arrive at empirical results. According to the study findings, the identified CGeoPOFDs offered traditional characteristics of network public opinion while also featuring unique spatiotemporal characteristics. Over time, CGeoPOFDs demonstrated spatial aggregation and bias diffusion and an overall positive emotional tendency.

## 1. Introduction

In recent years, unusual changes in the global climate have increased the incidence of floods. Meanwhile, the lack of valid protective measures has led to serious negative impacts on society [1,2]. In order to grasp the impact of floods in a timely and effective manner and facilitate effective responses to natural disasters, relevant agencies must obtain accurate public opinion information quickly. The rapid development of network information technology has meant that massive amounts of data focusing on flood disasters are generated in real time throughout the life cycle of such a disaster. These data often obfuscate disaster information and location information. Thus, analyzing these data can uncover the reflection of flood disasters at the social level, referred to as network public opinion [3].

The currently available literature on network public opinion can be roughly divided into three categories:

The first group of researchers concentrated on the network structure and dissemination mode of public opinion from the perspectives of time and content. These investigations abstracted each facet of public opinion information into network nodes and used social network analysis methods to analyze the characteristics and dissemination mode of network public opinion during emergencies [4]. Examples from this approach include opinion leader identification [5], tipping point analysis, and evaluating optimal topic classification [6,7]. However, most of the studies mentioned here focused on the temporal dimension, while investigations examining the spatial dimension remain scarce. In addition, these scholars’ findings tend to reflect the characteristics of network public opinion in terms of content, neglecting any consideration of flood disasters.

Studies that fall into the second category center around optimization strategies for the algorithms considered—specifically, optimizing the relevant algorithms by combining the structural characteristics of public opinion information with effective strategies. WEKA was introduced to compare the SVM classification and naïve Bayes classification [8]. Meanwhile, Ko used the negative class information of the text to optimize the naïve Bayes classification [9]. Multiclass boosting with adaptive group-based KNN was applicated in text classification [10]. Most of these optimization strategies for text classification algorithms were developed from a single point of view [11] and did not fully consider the position of feature items in the text, the semantic information of feature items, the structural features of text, and the impact exerted by the limitations of a single algorithm.

In the third category of studies, the researchers identified the location information from social media data and verified its validity in disaster analysis. At present, studies identifying hidden location information in social media data can be roughly divided into four approaches: dictionary-based, rule-based, machine learning-based, and deep learning-based. A model was used to extract computational representations of Chinese addresses [12]. This solution made it feasible to form certain structural rules according to the element features, part-of-speech features, and syntactic features of Chinese place names. On the basis of this possibility, the recognition of the location information can be translated into the problem of serialization, and random field is an excellent tool to handle the serialization problem. Then, the authors used the random field model to identify the location information in the text [13,14,15]. In order to make full use of the identified location information, relevant researchers further explored the real events behind the social networks. Information from Twitter was used to analyze super typhoon Haiyan [16]. Qu et al. used microblog data to explore disaster situations of the 2010 Yushu earthquake [17]. Choi et al. (2015) discovered a relationship between Twitter users’ geographic locations and the evolution of disaster situations from research on social big data [18]. Zhang and Cheng described spatiotemporal evolution and influencing factors of public sentiment in natural disasters [19]. Based on Hurricane Sandy, Neppalli et al. used Twitter data to analyze hurricane-related emotions and visualized the results [20]. These studies all focused on research that centered around location information and failed to integrate geography and social psychology into the investigation in any comprehensive way. Furthermore, these prior studies lack any consideration of the evolution law of public opinion.

The above analysis shows that most of the current research on network public opinion of disasters was carried out from a single perspective, such as the characteristics of the text, algorithm optimization, or spatial location recognition, and the studies rarely identify and analyze public opinion from multiple angles (e.g., spatiotemporal dimension of public opinion and potential social psychology). Nevertheless, a connection exists between the geospatial environment and social psychology. In recent years, researchers in this field have begun to pay attention to how people’s geospatial environment at different geographical scales can affect their psychology, along with considering the degree of influence. For example, PLAUT et al. found that people living in New England and the western mountainous regions of the United Kingdom had higher autonomy, while those in the southeastern and central regions were lower [21]. Relevant studies have also proved that social-psychological characteristics can reflect the characteristics of the geospatial environment, and scholars have observed differences in social-psychological characteristics at different geographical scales [22]. In addition, social-psychological characteristics have been shown to impact the generation and development of specific public opinion events [22]. In this context, Lai et al. proposed the concept of geographical public opinions [23]. This concept involves taking the spatial dimension as the starting point and combining big data technologies and psychological differences between different geographical regions in analyzing network public opinions. Accordingly, this study used spatial dimension and social psychology to examine public opinions of flood disasters. The research was based on the following hypothesis: When public opinions break out, the differences of network public opinions in geography will form spatially different centers of geographical public opinions in flood disasters (CGeoPOFDs). These centers represent the cities that network public opinion pays attention to. Based on this hypothesis, we collected microblog data from 29 June 2020 to 16 July 2020 and applied two optimization methods to optimize the naïve Bayes network classification algorithm. The Bayes network has been used to analyze flood disasters [24]. In order to include more semantic information in the feature items, we introduced syntactic parsing to improve the selection method for feature items in the naïve Bayes network classification algorithm. The single feature word was replaced by a feature word pair, and the multi-classifier fusion optimization was realized by using an ensemble learning strategy, whereupon the different weak classifiers were fused to generate a better classifier. Next, the optimized algorithm was used to filter out data. These data were related to flooding and reflected public opinion. Based on these data, CGeoPOFDs were identified and analyzed through geographic mapping, hotspot analysis, and sentiment analysis. Finally, the feasibility of the proposed method was verified by combining microblog social data and real-world disaster data.

The method proposed in this research allowed us to analyze network public opinion by integrating geography and social psychology to form CGeoPOFDs. According to the final results, each CGeoPOFD displayed the traditional characteristics of network public opinion while also featuring unique spatiotemporal characteristics. Over time, CGeoPOFDs evidenced spatial aggregation and bias diffusion and an overall positive emotional tendency. Another goal of this research was to reveal areas of public opinion with hidden dangers. By paying attention to different types of public opinion in different regions, different degrees of guidance can be adopted to achieve precise direction [25].

## 2. Methods

### 2.1. Naïve Bayes Network

Text classification is an important work in the field of text mining [10]. Scholars have employed many methods for text classification, such as decision trees, naïve Bayes networks, and neural networks.

Relevant studies have shown that KNN has a large cost of computation when dealing with a large-scale dataset [26]. The decision tree algorithm will not only be affected by singular data, but the efficiency of this algorithm will also be reduced with the increase in the amount of data. Although the accuracy of SVM is high, Wegener D points out that SVM is slow, and the cost of time is large when training a large dataset [27]. The naïve Bayes network has high adaptability and stable efficiency for text classification, and it is easy to obtain the parameters required by the algorithm. However, its classification accuracy is poor, which is also the problem to be solved in this research.

The naïve Bayes approach used in this study involved a simplified version of the Bayes algorithm [28], which assumed that the feature attributes were conditionally independent of each other when a target was given. Using the given datasets comprising the training sample, we generated the probability distribution from input to output. Thus, we applied the learning model to obtain an output *y* with the largest posterior probability corresponding to the input *x* [29,30].

The following conventions were applied for symbols used in our calculations: uppercase letters represent variables, lowercase letters denote variable values, *C* signifies categorical variables, and *X* is used for datasets to be classified.

According to the relevant theories, the text data to be classified belonged to the category with the largest posterior probability *P (c_i_|x)*, which we defined as follows:(1)$$P\left(c_{i}|x\right)=\frac{P\left(c_{i}\right)*P\;(x|c_{i})}{P\;\left(x\right)}$$
where *x* represents the text to be classified, and *c_i_* represents the category.

Our samples comprised microblog data. After preprocessing, the word segmentation set of each sample was used as the feature word vector. Accordingly, the input, *X* = {*x*_1_*, x*_2_*,* …, *x_n_*}, of the classification work was a collection of *n*-dimensional feature word vectors. The process assumed that the microblog data could be classified into two categories: opinion data and garbage data. Meanwhile, the output was the marked set of category *Y* = {*c*_1_*, c*_2_}. The probability that the text vectors belonged to class *c_i_* (*i* = 1, 2) was defined as follows:(2)$$P\left(\frac{Y=c_{i}}{X=x}\right)=\frac{P\left(\frac{X=x}{Y=c_{i}}\right)P\;\left(Y=c_{i}\right)}{\sum_{i=1}^{i=2}P\left(\frac{X=x}{Y=c_{i}}\right)P\;\left(Y=c_{i}\right)}$$

During the calculation process, if a category did not have a corresponding classifier in the training set, the final probability value was revealed to be 0. Because this outcome was contrary to the fact, we adopted the Laplace correction method for smoothing.

### 2.2. Optimization of Feature Word Vectors Selection

Natural language processing (NLP) is a field that facilitates analyzing Chinese text [31], including the recognition and extraction of Chinese text. Many useful methods have been developed that can be used for this purpose, such as syntactic parsing, semantic distance, and label propagation [32]. The current investigation used the syntactic parsing method [33] to optimize the judgment of feature word vectors in a naïve Bayes network. Specifically, we replaced a single feature word with a more semantically informative feature word pair. The optimized classifier was named W2_NB.

The analyzer used in this research came from the Language Technology Platform (LTP) shared package of the Harbin Institute of Technology. For Chinese text, twenty-four deterministic dependency types and one uncertain relationship (Table 1) were established.

Each word in a particular sentence was considered to be a child element. Therefore, we added an analysis of the nature of the words in performing syntactic parsing. Table 2 provides a list of the tagging set representing the nature of various types of words. This set was promulgated in the 2003 National 863 Analysis and the evaluation of tagging.

We marked the following word types as opinion words: n, ns, r, v, a, z, d, i. In addition, we selected SBV, ATT, ADV, CMP, and VOB as the basic dependencies. On the basis of these relationships, we established rules for some combinations: SBV+ADV, ATT+SBV+ADV, SBV+VOB, ATT+SBV+VOB, and SBV+CMP. The distance between words in the Chinese text also affected the strength of the relationship between words. Specifically, Liu found that the dependency distance between words in Chinese was 2.81 [34]. Therefore, we set a distance of three units as the threshold for extracting word pairs. Table 3 presents an example showing a piece of misclassified data in the source data. The table displays the basic information contained in these data, and the username was processed.

First, the text was segmented to obtain the result in Chinese: “杭州”, “的”, “雨”, “下”, “太久”, “了”, “心情”, “很”, “不好”. Tagging of parts of speech was performed on each segmented word, yielding the following results: ns, u, n, v, d, u, n, d, and a. The syntactic parsing result of this case is shown in Figure 1.

According to the results of syntactic parsing, the vectors that we needed were “心情/很/不好” (I am in a bad mood), corresponding to “SBV+ADV,” and “雨/下/太久” (It has been raining for too long), corresponding to “SBV+CMP.” Therefore, in the following classification work, these word pairs were combined as the feature word pair vector for this piece of data. Furthermore, this vector was brought into the subsequent algorithm.

### 2.3. Optimization of Multi-Classifier Fusion

Ensemble learning, in which multiple learners are combined to complete specified learning tasks, can generally be divided into three categories: bagging, boosting, and stacking. Bagging, also known as resampling with replacement, requires that after sampling the total sample data, the collected samples must be returned to the initial set, and the process will be repeated. Boosting is a strongly dependent integration strategy. This category requires that a base learner is obtained according to the initial training set, and the attention of the training samples can be increased based on the learning results. Next, a new base learner will be obtained based on the adjusted training samples. This process is done iteratively until the iterative termination conditions are met: the number of base learners meets a certain requirement, or the integration effect fulfills a certain requirement. Finally, multiple base learners are weighted and combined to obtain the final learner. The main working mechanism of stacking is the use of a meta-classifier or a meta-regressor to integrate multiple classification models or regression models. The entire training set is used as input, while the output of the base model is used as the input to train the meta-model, and a complete model is built.

After considering the characteristics of each ensemble strategy, we chose to apply a multi-classifier fusion optimization based on the stacking ensemble method that could be used to improve the prediction results. In order to make full use of the characteristics of the two algorithms, NB and W2_NB, we applied a heterogeneous ensemble method to fuse these two algorithms. This method entailed deriving meta-learners from different algorithms. In order to improve the generalization ability of ensemble learning, the “voting method” was adopted to combine multiple prediction results and thereby reduce the dependence of multiple meta-learners. The flow of the multi-classifier fusion optimization strategy proposed in this research can be described as follows (See Algorithm 1).

**Algorithm 1:** The flow of the multi-classifier fusion optimization strategy
**Input:**
Initial training set: *S_0_={(x*_1_*,y*_1_*),(x*_2_*,y*_2_*),…,(x_n_,y_n_)}*;Initial learning algorithm: *A*_1_ = NB, *A*_2_ = W2_NB;Combination strategy: *V*.
**Process:**
  for *t* = 1, 2 do    *L*_t_ = *A_t_*(*S_0_*);  end for  *S’* = Ø;  for *i*   = 1, 2, …, m do    for *t* = 1, 2 do      *Z_it_* = *L_t_*(*x_i_*);    end for   *S*’ = *S*’ ∪ ((*Z_i1_, Z_i2_*), *y_i_*);  end for  *L*’ = *V(S’)*
**Output:**
*H*(*x*) = *L’*(*L*_1_(*x*)*, L*_2_(*x*))

Integrating multiple weak classifiers can often result in obtaining a strong classifier. The classifier obtained in this study was defined as “S_W2_NB.”

### 2.4. Geographic Mapping

In order to map the text data to the specified geographic area, we proposed a geographic mapping method based on the optimization of the “relative majority voting method”.

We crawled the representative POI (points of interest) data from cities across the country. After deduplication, the number of POIs was 821,727. Based on these data, the city feature vector *T_j_* = {*POI_j_*_1_*, POI_j_*_2_*,* …, *POI_jm_*} was constructed. *T_j_* represented the city name, while {*POI_j_*_1_*, POI_j_*_2_*,* …, *POI_jm_*} indicated the POI information corresponding to the city *T_j_*. For example, the following city feature vector might be constructed for “Nanjing”: Nanjing = {Nanjing, Xinjiekou, …, Confucius Temple} using the chosen method based on the “relative majority voting method.” When POI information appeared in a piece of text data, the vote of the text was increased by one. If the text data had multiple cities with the highest vote, then we further considered the vote of these cities in all of the text of the day and selected the city with the highest vote. If there were still multiple cities with the highest vote, we randomly selected one as the city to which the current text data belonged. Finally, the corresponding latitude and longitude information according to the city was obtained, and these text data were abstracted into points according to the latitude and longitude.

### 2.5. Hotspot Analysis

Hotspot analysis refers to the statistical calculation of each piece of data using the General G index. The spatial location of data clusters with a higher or lower value could be determined according to the statistical results. In this analysis, we needed to consider a single spatial feature and multiple features within its vicinity. Therefore, a single high value was not seen as a hotspot; instead, only when a feature and its neighboring features had high values was this area defined as a high value or a high-value cluster, that is, a hotspot. Conversely, when a feature and its neighboring features were low-valued, it was defined as a low value or a low-value cluster.

The hotspots in the network public opinions were identified as CGeoPOFDs. In order to identify these centers, the points were mapped to the prefecture-level city map. Taking the prefecture-level cities of China as the research object, we took the number of points in each map spot as the quantitative attribute, and then completed the hotspot analysis.

### 2.6. Sentiment Analysis

In order to describe the bias of public opinions in each hotspot area, we conducted sentiment analysis wherein the overall emotional state of the current hotspot area was used to represent the bias of public opinions. The SnowNLP, a function library of Python, was used to construct and train the determination model for the sentiment index while considering that the characteristics of the junk data from microblogs would change over time. Therefore, when creating a self-built sample database, we adopted the practice of randomly sampling samples throughout the entire period, which meant randomly selecting samples from all data from each day. As a result, 500 samples from each day were added to the self-built sample library.

In order to further describe the trend of overall public opinions, we constructed two indexes for expressing emotion based on the emotion index of each test, which we labeled the Positive Tendency Index (*P_tend_*) and Negative Tendency Index (*N_tend_*). These indexes were defined as follows:(3)$$P_{tend}=\frac{\sum_{i=1}^{P_{count}}\;\left(P_{index}-0.5\right)}{\;\left(\sum_{i=1}^{P_{count}}\;\left(P_{index}-0.5\right)+\sum_{j=1}^{N_{count}}\;\left(0.5-N_{index}\right)\right)}$$
(4)$$N_{tend}=1-P_{tend}$$
where *P_count_* represented the number of data whose sentiment index was greater than or equal to 0.5, and *N_count_* denoted the number of data whose sentiment index was less than 0.5. Lastly, *P_index_* indicated the sentiment index of each piece of emotionally positive text, while *N_index_* signified the negative.

### 2.7. Research Process

Based on related methods, the following feasible recognition process was proposed (Figure 2).

First, from the relevant literature, we selected ten keywords in Chinese that were strongly related to flood disasters: 暴雨(rainstorm), 排涝(drainage), 山洪暴发(flash flood), 洪灾(flood), 泛滥成灾(flooding), 防汛(flood control), 泄洪(flood discharge), 抢险(emergency rescue), 泥石流(mudslide), and 台风(typhoon). These keywords were expanded through synonyms, a function library of Python. Next, one hundred and five keywords were obtained for the search of microblog data. We collected microblog data from 29 June 2020 to 16 July 2020. The number of this dataset was 2,030,967. Then, we preprocessed this dataset. We did some work to remove redundancy, such as removing data without content, removing duplicate data, and so on. Since we need the location information of the text, we identified the location of the text and eliminated the text without location information. After preprocessing, the number of the dataset was 511,013. Then, the preprocessed microblog data were brought into three classification models for comparison. The best result was selected to establish the opinion dataset. This dataset, administrative division data, and POI were combined for geographic mapping to obtain an opinion dataset with a location. This dataset was input into the sentiment index calculation model in order to add sentiment index information to each text. The final tests were then mapped to each geographic patch, and the number of opinion data items in each geographic patch was used for hotspot analysis to obtain the final centers of geographic public opinions in a flood disaster. Finally, three types of analysis were conducted based on the analytical results and real-world data concerning Poyang Lake flooding.

## 3. Results

### 3.1. Comparing Classification Algorithms

We used the NB, the W2_NB, and the S_W2_NB for our experiments. As shown in Section 2.1, the microblog data had been classified into two categories: opinion data and garbage data. The label, opinion data, mean that these data contained opinions. The garbage data did not contain opinions. In the experiment, a comparative analysis was carried out from dimensions featuring seven different numbers of samples (1000, 2000, 3000, 4000, 5000, 6000, 7000). Considering the balance of the dataset, we ensured the same proportion of the dataset with each label when collecting samples. In each dimension, we split the samples in a ratio of 7:3, with training data accounting for 70% and test data accounting for 30%. We adopted a random sampling method to ensure that the proportion of the dataset with different labels in the training dataset was the same.

Three indicators (precision, recall, F-score; see Table 4) and a graphical representation of the F1 (Figure 3) were obtained. In general, comparing the accuracy of different classification algorithms involves averaging different indicators (micro average and macro average). For this experiment, we selected the macro average method to calculate the average value of different indexes of different algorithms under different categories. Based on the calculation result, the average of the index was used to represent the effect of the algorithm.

As can be seen from Table 4, the precision, recall, and F1 of the traditional naïve Bayes classification algorithm were generally low, and the classification performance was the worst. Compared with the W2_NB and the NB, the S_W2_NB showed an improvement in the classification effect. Three indicators of the S_W2_NB algorithm were significantly improved, and the classification effect was the best. Then, we made the confusion matrix according to the result of the S_W2_NB algorithm (Appendix A).

In order to verify the robustness of the different algorithms, we conducted our analysis from seven dimensions of the number of samples. Since the F1 was a comprehensive indicator that was calculated based on precision and recall, we constructed a line graph of the F1 under different dimensions (Figure 3). Figure 3 illustrates that the F1 of the NB and the W2_NB fluctuated and rose as the number of samples increased. In contrast, the S_W2_NB demonstrated obvious stability, indicating that the robustness of the S_W2_Nb was stronger than that of the others. Furthermore, the S_W2_NB obtained the highest F1 (0.878) when the number of samples was 3000. Therefore, the model under this dimension was used for later text classification works.

### 3.2. Analysis of Spatiotemporal Evolution of CGeoPOFDs

We analyzed microblog data from 29 June 2020 to 16 July 2020. The temporal evolution information (Figure 4) and the spatial evolution information (Figure 5) of CGeoPOFDs during this time period were obtained through related experimental methods.

As shown in Figure 4, as time went on, the number of CGeoPOFDs revealed the life cycle (generation → continuous warming → high tide → lower → regeneration). This outcome also proved that the results of the text classification algorithm (S_W2_NB) adopted in this experiment were authentic. According to the changes in the number of CGeoPOFDs, we divided the entire time period into three stages: the first stage (from July 1 to July 3), the second stage (from July 7 to July 11), and the third stage (from July 12 to July 16). The partial child life cycle was included in each stage. The child life cycle was in line with the characteristics of network public opinions in reality. In order to describe the hierarchy of CGeoPOFDs on different dates, we divided CGeoPOFDs vertically into three levels according to the high-value interval in hotspot analysis: strong CGeoPOFDs (with a confidence level of 99%), secondary CGeoPOFDs (with a confidence level of 95%), and low CGeoPOFDs (with a confidence level of 90%). From the perspective of the entire time series, the first and third stages were dominated by strong CGeoPOFDs, whereas the second stage was dominated by secondary CGeoPOFDs and low CGeoPOFDs.

Figure 5 shows the location of the affected areas. The light orange border indicates the affected provinces. In addition, the light green areas indicate all CGeoPOFDs. According to the illustrations in Figure 6, CGeoPOFDs were all located in the middle and lower reaches of the Yangtze River; furthermore, CGeoPOFDs at different stages showed the character of aggregation. Moreover, the aggregation areas of different stages shifted spatially over time. In these CGeoPOFDs, the hotspot city “Aba Tibetan and Qiang Autonomous Prefecture” in Sichuan Province, which appeared on July 3 and July 11, was an unusual hotspot. According to our analysis of the source data, due to the effects of the flood disasters, mudslides occurred in some areas on July 3, reaching their peak on July 6 and July 7. These mudslides were gradually brought under control, and the life cycle of public opinions concerning them ended on July 11. Compared with the disaster situation of the centers in the second stage, the situation regarding the mudslides was relatively weak. In addition, under the influence of “Habitual Psychology,” this autonomous prefecture did not evolve into a center in the early part of the second stage.

In the first stage, CGeoPOFDs spread out, with Wuhan, Hubei Province, as the gathering center. This situation involved a relationship with the location of the rain clouds in the early stage of the flood disaster. In the second and third stages, CGeoPOFDs all spread out, with Poyang Lake in northern Jiangxi Province as the center. Comparing the area of the two stages mentioned here revealed that CGeoPOFDs in the second stage spread from Poyang Lake to the lower reaches of the Yangtze River. Some cities in southern Anhui Province, some cities in central and southern Jiangsu Province, and some cities in northeastern Zhejiang Province were included in this diffusion area. In addition, the level of CGeoPOFDs roughly decreased, moving outward from Poyang Lake. In the third stage, the clustering area of CGeoPOFDs returned to the middle reaches of the Yangtze River and spread from Poyang Lake to the northwest. However, compared with the level of CGeoPOFDs in the second stage, the centers in the third stage were generally secondary CGeoPOFDs.

### 3.3. Analysis of Temporal Evolution of Sentiment Index

We conducted a statistical analysis of the sentiment analysis of CGeoPOFDs in different stages. Since the overall standard deviation of the sentiment index on different days (all in the range of 0.301–0.367) was small, we chose the average as the analysis indicator.

On the whole, most of the sentiment indexes were greater than 0.5, except on July 7 and July 15. This outcome suggests that, in the case of flood disasters, the network environment of CGeoPOFDs is good, and the emotions of netizens are positive. In different stages, the change of emotional index shows inverted U-shaped and left inverted U-shaped trends. At the beginning and end of each stage, the sentiment index was lower, and the indices for July 7 and July 15 were also included.

In the initial stage of public opinion (the first stage), the sentiment indexes were all greater than 0.5. This situation was related to the severity of the flood disaster at that time. As the flooding grew in severity, some emergencies not only caused the child life cycle of many CGeoPOFDs but also made the sentiment index at the beginning of the life cycle low and even negative.

## 4. Discussion

This investigation used a naïve Bayes network algorithm as the basic classification algorithm and incorporated two particular strategies (syntactic parsing and ensemble learning) to improve it. According to the relevant indicators (Table 4, Figure 3), it can be seen that the two optimization strategies in this research had an effect in terms of classification accuracy, and “S_W2_NB” showed the best robustness. The classification efficiency of naïve Bayes was found to be relatively stable. However, because of the difficulty inherent in satisfying its independence assumption, the classification accuracy was easily affected by feature items. Therefore, methods for improving this algorithm can be divided as follows: (a) reducing the constraints of independence, such as tree-augmented naïve Bayes algorithms, or (b) analyzing the correlation between the feature items to find the feature items with greater influence, such as semi-naïve Bayes algorithms. It is also necessary to consider the many feature items in a text, along with the existing optimization strategies needed to process the feature items, which tend to increase time complexity and reduce classification efficiency. Due to the noise feature items in Chinese text, we applied syntactic parsing to extract representative word pairs and used the latter as the feature items, which reduced the time complexity involved in the process. A comparison of the classification results of the NB and the W2_NB reveals that the NB mistakenly classified a part of the opinion data as garbage data, while the W2_NB avoided this situation. This difference can be explained in terms of the W2_NB improving the judging method of feature word vectors in the NB. A simple set of single words was no longer used for feature items; instead, the feature items used comprised word pairs with more semantics and better generalization of the sentence meaning. However, the optimization effect achieved by the W2_NB was not good, as the F1 was improved by only 0.067, on average. This outcome resulted because the number of feature items obtained by the W2_NB was small, and the word pairs with an opinion under the same period and topic were similar. Although these key feature items could express most of the semantics, they suffered the disadvantage of insufficient category coverage. Consequently, we adopted the idea of ensemble learning by taking the NB and the W2_NB as the base classifiers and making full use of the two algorithms to construct a strong classifier (S_W2_NB). Compared with the NB and the W2_NB, the F1 of the S_W2_NB was respectively improved by 0.237 and 0.170, on average.

The CGeoPOFDs identified in this research had spatiotemporal characteristics. Table 5 provides a chronological presentation of the main information about this disaster.

In the whole time series, the number of CGeoPOFDs exhibited a macro and micro life cycle, and there were differences in the level of cities at different stages. This observation reveals temporal heterogeneity in CGeoPOFDs (Figure 4). This outcome is in line with the characteristics of the network public opinions while also proving the reliability of the method for identifying the centers used in this research. A relationship emerged between the temporal heterogeneity of the centers and the actual flood disasters. For example, in the first stage, the severity of floods was low, and the spatial location of the rain clouds resulted in fewer CGeoPOFDs. Disaster information from July 4 to July 6 led to the first sharp increase in the number of hotspot cities during the second stage, which continued to increase. Next, the collapse of the embankment on July 9 led to the second outbreak during this stage. In addition, there were differences in the level of CGeoPOFDs at different stages. For example, the first stage was dominated by the strong category, the second stage was dominated by the strong and the secondary, and the third stage was dominated by the secondary and the low. Analysis of the public opinions source data revealed that, because the cities around Poyang Lake continued to be the center, people had the “habitual psychology” about the flood disasters of this area. For other areas or in the early stage of the disaster, people would have been sensitive to sudden flood events, resulting in many strong CGeoPOFDs. This phenomenon supports the rationality of the hypothesis of this research.

Our analysis of the spatial evolution of CGeoPOFDs showed that, in the entire public opinions cycle, the clustering area of CGeoPOFDs shifted from the middle and upper reaches of the Yangtze River to the middle and lower reaches of the Yangtze River and, finally, back to the middle and upper reaches of the Yangtze River. During this process of the deviation of the clustering center, the phenomenon of bias diffusion was formed, meaning that CGeoPOFDs spread from the clustering center to the surrounding areas, and the level of CGeoPOFDs decreased from the clustering center to the outside. The diffusion direction was biased toward the offset direction of the clustering center, while a new clustering center of CGeoPOFDs was formed within its diffusion area. The above evolution trend suggests that CGeoPOFDs can be characterized by clustering and bias diffusion. This phenomenon may be intrinsically related to the “habitual psychology” mentioned earlier.

While identifying the centers, in order to describe the public opinions situation of these centers, we used multiple indicators to construct a sentiment index and analyzed the temporal evolution of sentiment (Figure 7). As a whole, during this disaster, netizens’ sentiment toward the centers was positive, and the three stages were characterized by inverted U-shaped and left inverted U-shaped trends. Additionally, the beginning and the end of each stage represented the lowest point of the sentiment index. Furthermore, two abnormal points (where the sentiment index was less than 0.5) were observed: the sentiment index on July 7 was 0.481, while the sentiment index on July 15 was 0.462. Combining this information with Table 5 reveals that the events on July 7 and people’s psychology toward emergencies made the sentiment index less than 0.5 on that day. The official announcement about the disaster on July 15 resulted in the sentiment index reaching less than 0.5 on that day.

In order to show the information concerning the daily network public opinions, the hotspot word-based information of the day was counted to represent the overall trend of public opinions (Figure 8).

As time passed, the overall public opinions were dominated by hotspot words such as “heavy rain” and “flood disasters”, along with sub-hot words such as “police”, “college entrance examination”, and “Changsha”, which appeared intermittently. This observation shows that throughout the entire cycle, the public opinions brought about by the flood disasters were always in a dominant position, and keywords such as “police” indicated the societal response to the flood. Our analysis of the overall trend of public opinions uncovered some sudden and short-lived hotspot events. For example, keywords such as “Changsha” that appeared on July 12 rose to become the most popular keywords on July 13 and then went down and disappeared on July 14. Our analysis of the source data showed that, during this time period, the flood disasters caused the collapse of a crane in Changsha. This incident was properly handled on July 14. Analysis of the overall trend from the dimension of the sentiment index highlighted the sentiment tendency of the same hotspot events in the current period. For example, the hotspot words related to the “crane” were all in the negative area, demonstrating that this event presented a negative emotional tendency on the internet. Meanwhile, “gaokao” appeared in two areas, and the hotspot words in the positive area occupied a higher position, which indicates that this hotspot word represents a positive emotional tendency. Therefore, this method can be used in potential emergencies to find the emotional tendency of emergencies and their evolution.

This investigation identified and analyzed CGeoPOFDs, proving the feasibility of integrating spatial dimension and social psychology into the research of public opinions in flood disasters. Furthermore, we were able to verify the feasibility and necessity of identifying CGeoPOFDs. Drawing from the traditional theoretical methods of disaster investigations, we proposed and verified the hypothesis that flood disasters can be characterized as creating centers of geographical public opinions. We also provided a reference direction for this research focused on network public opinions by including the background of the disasters.

In addition, the methods used in this study to identify and analyze the centers may also be applied to other types of network public opinions. Examples of investigative areas where this concept may be helpful include the centers of the geographic public opinions against the background of the new coronavirus pneumonia (Corona Virus Disease 2019, COVID-19) and the centers of the geographic public opinions against the background of policy implementation. Through the identification and analysis of the centers, relevant government agencies and organizations can implement precise, advanced policies.

The geographic mapping method used in this research was an optimization of the “relative majority voting method”. Nevertheless, this method had defects. The correlation between the text to be mapped and POI might have been insufficient, or they might have been indirectly related. These defects led to the inaccurate geographic mapping of text, such as “Aba Tibetan and Qiang Autonomous Prefecture”. As a result of our analysis, we understand that it is difficult to identify a potential location using the geographic mapping method applied in this research. Furthermore, this method uses the vote from other regions on the same day as a reference standard. If the vote in other regions on the same day did not strongly differ from the number in the autonomous prefecture, the latter did not evolve into the center. However, the processing and analysis of massive data can reduce the errors caused by this defect, as has been proved by the final results of this research. In the future, we will carry out additional research based on this investigation to obtain a more comprehensive understanding of the correlation between the text to be mapped and POIs. In addition, we plan to build a fast, effective geographic mapping method and consequently improve the efficiency and accuracy of identifying CGeoPOFDs. Lastly, based on the results of this research, another potential direction for further research involves discovering how to quantitatively express the correlation between CGeoPOFDs and actual disasters.

## 5. Conclusions

The “S_W2_NB” proposed in this research has been shown to improve the effectiveness of text classification and the robustness of the algorithm. Since the independence assumption of naïve Bayes was difficult to satisfy, the text classification effect based on this algorithm was poor. Due to the small number of feature items extracted by W2_NB and the existence of similar opinion word pairs, the final optimization effect was not good (the F1 increased by 0.067, on average). However, S_W2_NB combined with the advantages of the NB and the W2_NB obtained a higher optimization effect (the F1 increased by an average of 0.237 and 0.170, respectively) and was more robust.

CGeoPOFDs have the feature of clustering and the feature of bias diffusion. In the time dimension, the CGeoPOFDs that we studied showed a macro and micro life cycle. Furthermore, the influence of “habitual psychology” led to differences in the distribution of CGeoPOFDs. In the context of public opinions in flood disasters, the spatiotemporal evolution trends of CGeoPOFDs included some conventional public opinion characteristics, a phenomenon that proved the authenticity and reliability of this CGeoPOFDs method.

During the entire flood disaster period, the sentiment index showed a positive trend overall. However, lower points could be observed at the beginning and end of each sub-life cycle (0.481 on July 7 and 0.462 on July 15). Over time, the centers demonstrated the following phenomenon: “mainly high-hot events, with sub-hot events appearing intermittently”, which was closely related to the overall public opinions and emergencies that occurred in the current stage. In conclusion, based on our findings, this phenomenon can be applied in analyzing public opinions and properly guiding disaster response efforts.

However, we found some Limitations and Challenges in this research. The geographic mapping method used in this research had defects. These defects may lead to the inaccurate geographic mapping of text. This is a problem that needs to be solved properly. In addition, we need to optimize the model to achieve lower time complexity. In the future, we will build a fast and effective model to find the location information of text. Then, we will carry out research on quantitatively expressing the correlation between CGeoPOFDs and actual disasters.

## Figures and Tables

**Figure 1 ijerph-19-10809-f001:**
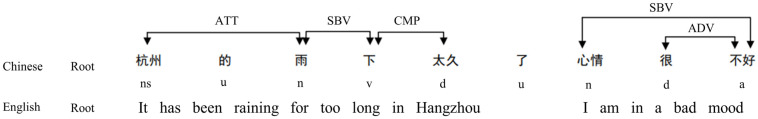
The syntactic analysis of the case.

**Figure 2 ijerph-19-10809-f002:**
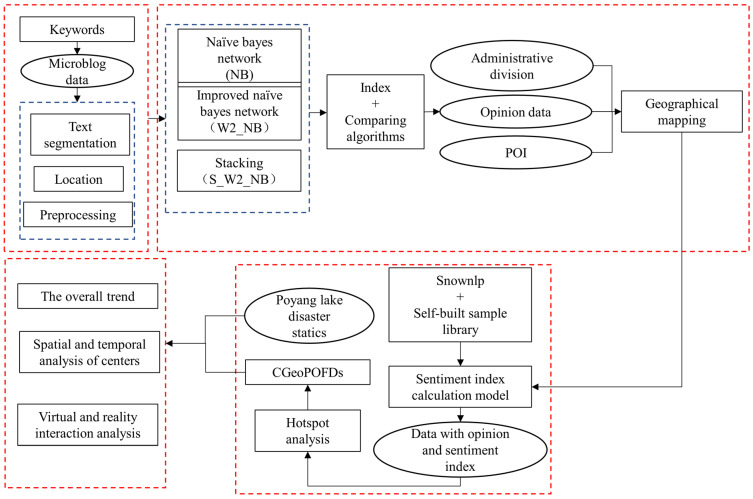
The model used to identify and analyze CGeoPOFDs. The red boxes show the different steps and reflect the inner processes.

**Figure 3 ijerph-19-10809-f003:**
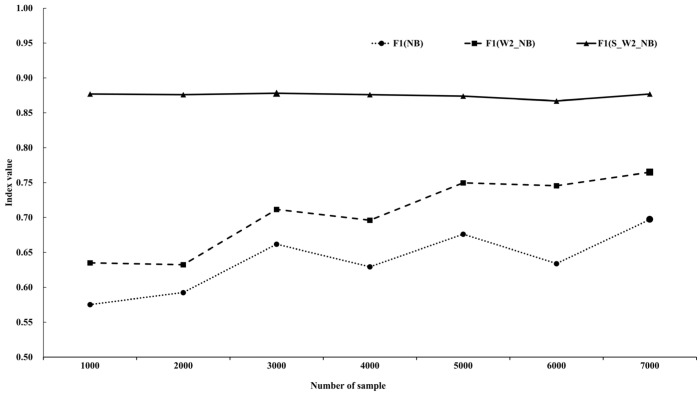
The index of comparison (F-score).

**Figure 4 ijerph-19-10809-f004:**
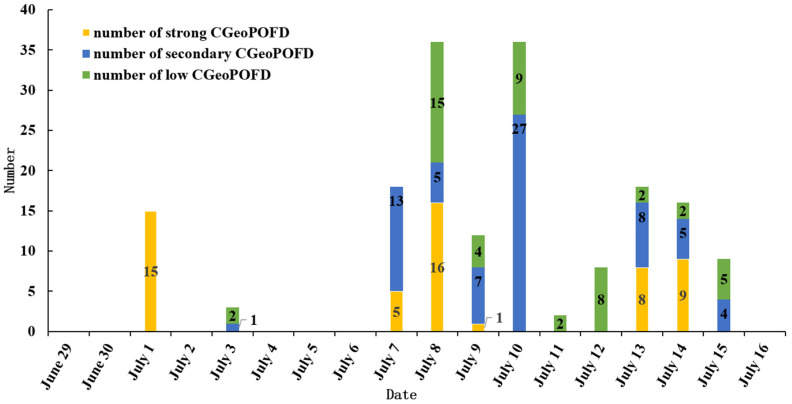
The temporal evolution of the number of CGeoPOFDs.

**Figure 5 ijerph-19-10809-f005:**
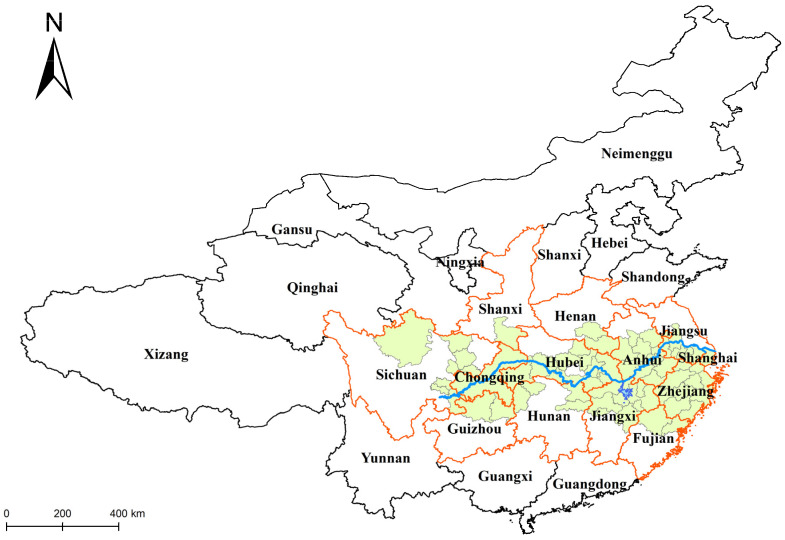
The location of the affected areas.

**Figure 6 ijerph-19-10809-f006:**
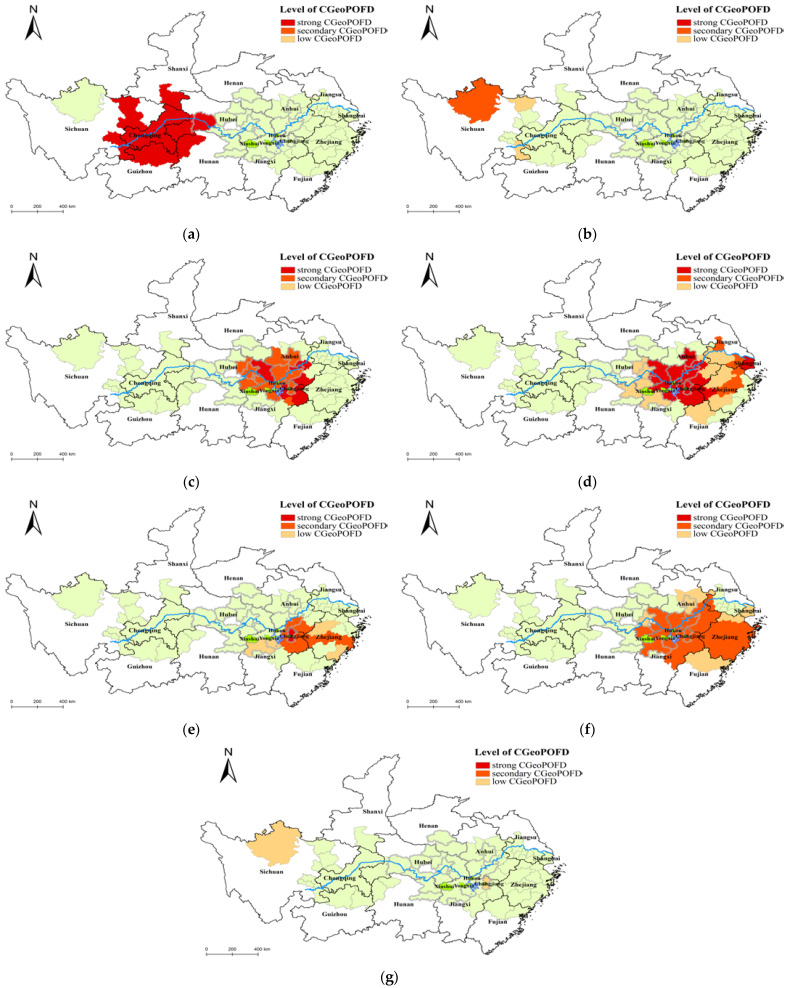
The temporal and spatial evolution of CGeoPOFDs. The first stage is shown in (**a**,**b**). The second stage is depicted in (**c**–**g**). The third stage includes (**h**–**k**).

**Figure 7 ijerph-19-10809-f007:**
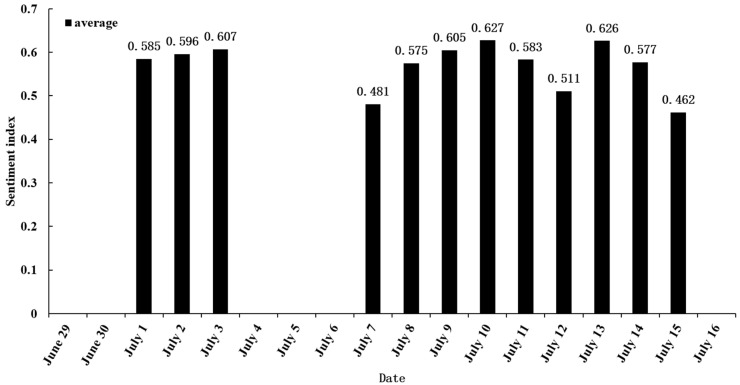
The temporal evolution of mood index.

**Figure 8 ijerph-19-10809-f008:**
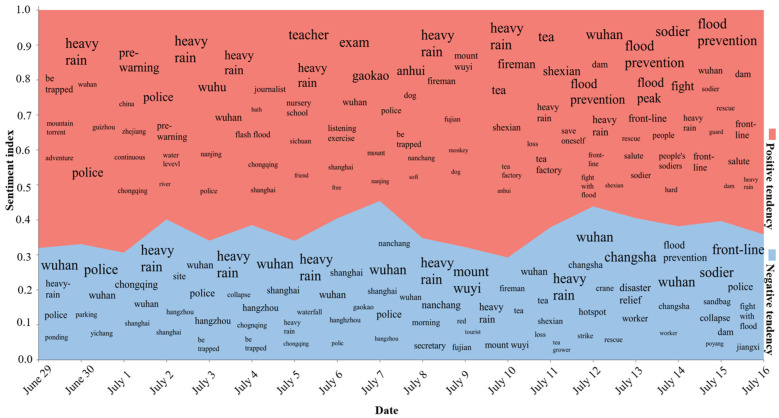
The overall trend of public opinions.

**Table 1 ijerph-19-10809-t001:** Type of relationship for dependency parsing.

Symbol	Implication	Symbol	Implication	Symbol	Implication
ATT	attribute	DI	adverbial	SBV	subject-verb
QUN	quantity	BA	put the object before the verb	HED	head
COO	coordinate	BEI	passive structure	MT	mood-tense
RAD	right adjunct	NOT	uncertain relationship	ADV	adverbial
POB	propositions-objects	IC	independent clause	DE	substitute object
SIM	similarity	VV	continuous verb structure	DEI	degree or condition
CNJ	conjunctive	APP	appositive	DC	dependent clause
IS	independent structure	LAD	left adjunct	—	—
CMP	complement	VOB	verb-object	—	—

**Table 2 ijerph-19-10809-t002:** The 863 POS set.

Symbol	Implication	Symbol	Implication	Symbol	Implication
n	none	c	conjunction	nt	time none
nd	locative none	j	abbreviation	v	verb
d	adverb	ni	institution name	e	interjection
k	subsequent element	b	distinguishing words	wp	punctuation
nh	name	u	auxiliary words	nl	place none
m	numeral	g	morpheme words	a	adjective
p	preposition	nz	other proper noun	h	preceding element
i	idiom	r	pronouns	ws	string
ns	place name	o	onomatopoeia	z	state words
q	quantifier	x	non-morpheme word	—	—

**Table 3 ijerph-19-10809-t003:** One case of microblog data.

	Username	User Type	Text	Time
Chinese	萧萧	微博会员	杭州的雨下太久了心情很不好	2020-07-06
English	Xiaoxiao	Microblog member	It has been raining for too long in Hangzhou, I am in a bad mood.	6 July 2020

**Table 4 ijerph-19-10809-t004:** The comparison of classification algorithms.

Count	NB			W2_NB			S_W2_NB		
	Precision	Recall	F1	Precision	Recall	F1	Precision	Recall	F1
1000	0.679	0.499	0.575	0.657	0.614	0.635	0.872	0.883	0.877
2000	0.691	0.519	0.593	0.648	0.617	0.632	0.870	0.883	0.876
3000	0.718	0.614	0.662	0.699	0.725	0.712	0.875	0.881	0.878 *
4000	0.686	0.581	0.629	0.683	0.710	0.696	0.873	0.880	0.876
5000	0.700	0.654	0.676	0.727	0.775	0.750	0.875	0.873	0.874
6000	0.676	0.597	0.634	0.726	0.766	0.746	0.865	0.870	0.867
7000	0.719	0.677	0.697 *	0.742	0.790	0.765 *	0.870	0.885	0.877

“*” indicates the maximum value of the F1 of the corresponding algorithm.

**Table 5 ijerph-19-10809-t005:** Review of Poyang Lake flood events.

Date	Events	Date	Events
June 29	Heavy rain in northern Jiangxi Province.	July 8	Improved the flood control response level and disaster relief response level of Jiangxi Province.
June 30	Heavy rain in central and northeastern Jiangxi Province.	July 9	The embankment collapsed at around 21:00 in Zhongzhou Polder, Poyang County.
July 1	—	July 10	Improved the flood control response level and disaster relief response level of Jiangxi Province.
July 2	The water level of Poyang Lake Xingzi Station was 18.01m.	July 11	The water level of Poyang station in Raohe exceeded the historical extreme value in 1998.
July 3	Flood control emergency response launched in Jiangxi Province.	July 12	The triangular polder collapsed in Yongxiu County, Jiujiang City, Jiangxi Province.
July 4	Floods occurred in Changjiang and Xiuhe, and over-alarm flood occurred at Xiangzi Station.	July 13	The length of the rupture of the delta link spread, and 23,411 people were transferred.
July 5	The water level of Xingzi Station was 19.14m.	July 14	Flood red warning in Poyang Lake dropped to orange warning.
July 6	Yellow flood warning issued in Jiangxi Province.	July 15	More than 6.42 million people were notified of the disaster, and there were 1007 disaster-stricken points in the province.
July 7	Orange flood warning was issued in Jiangxi Province.	July 16	The gaps in triangulation were closed.

## Data Availability

The text data were crawled from microblogs by using a program written in Python. A program written in Python was used to obtain POI from AutoNavi’s interface. The actual disaster data were obtained through the relevant official notification. City-level zoning data were obtained from relevant departments.

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
