# Peer review of "The Identification and Analysis of the Centers of Geographical Public Opinions in Flood Disasters Based on Improved Naïve Bayes Network"

_ijerph, 2022, doi:10.3390/ijerph191710809_

Round 1

Reviewer 1 Report

1) The article entitled "The Identification and Analysis of the Centers of Geographical Public Opinions in Flood Disasters based on improved Naïve Bayes Network" presents an interesting look at the risk of flooding through the prism of Internet listening (network monitoring) and in relation to space and time (in a spatial reference). The authors used the relationship between space (natural and spatial conditions) and social psychology to examine public opinion on flood disasters.

2) The article is interesting and the authors analyzed the relationship between the geospatial environment and social psychology. The idea of spatial inference based on data extracted from the text of microblogs is very good and promising for further research. The article is well-structured, but in the introduction I suggest you to better emphasize the research gap.

3) Figure 5 shows some dependencies in the spatial relation. The names of the various provinces / municipalities are described, but where they are located is not shown. In my opinion, the article should be extended with a figure presenting the location of the research area in a wider context, i.e. in relation to the country, province, continent, etc.

4) What are the possibilities of automating the processes of obtaining information from text and assigning them locations in space? The question is about the method described in the article, but also about other methods. Please comment.

Author Response

Response to Reviewer 1 Comments

Many thanks to the editor/ reviewer for the constructive comments on the previously submitted paper (Manuscript ID: ijerph-1869937). Below is my reply to these comments.

Point 1: The article is interesting and the authors analyzed the relationship between the geospatial environment and social psychology. The idea of spatial inference based on data extracted from the text of microblogs is very good and promising for further research. The article is well-structured, but in the introduction I suggest you to better emphasize the research gap.

Response 1: Thank you for your suggestions. In the introduction, we wanted to classify the existing literature and drew out the focus of this research from the shortcomings of each category. Therefore, when introducing existing researches, we made comparative analysis from the perspective of categories. Writing in this way, we may not have paid more attention to the specific literature. Therefore, we add more specific descriptions to some of the literature. (Line 74- 79)

Point 2: Figure 5 shows some dependencies in the spatial relation. The names of the various provinces / municipalities are described, but where they are located is not shown. In my opinion, the article should be extended with a figure presenting the location of the research area in a wider context, i.e. in relation to the country, province, continent, etc.

Response 2: This is a very good suggestion. We had added Figure 5.

Point 3: What are the possibilities of automating the processes of obtaining information from text and assigning them locations in space? The question is about the method described in the article, but also about other methods. Please comment.

Response 3: Thanks to reviewer’s comments. We aim to propose a model framework that can be used to identify and analyze geographic public opinion centers. Moreover, obtaining information from the text and geographic mapping were very important parts, which were also the focus of this research. In recent years, the research of obtaining location information from Chinese text is a hot topic, and a variety of models have been proposed in related fields, such as random field mentioned in this research. At present, we can not only extract some shallow location information, but also extract deep semantic information. The method proposed in this research belongs to the category of supervised classification, and the results show that this method has certain feasibility. If we want to automate the process, we need to build more representative, more generalized samples and train them into models to improve the possibility of process automation. In addition, through the later optimization of this paper, this method will also have higher automation ability.

Reviewer 2 Report

Research Summary

This manuscript deals with an interesting problem for supporting civil defense agencies in supporting people affected by floods and other natural disasters since the applied methodology has universal ends. Machine learning, natural language processing methods, and consequently text mining approaches are powerful tools to support the identification of dangerous situations reported by social web users and evaluate, for instance, the risk level according to their sentiments, emotions, or opinions, and the decision-making in the civil defense agencies to act quickly, ensuring people’s safety. 

Major Strengths

1. The authors proposed improvements in a simple but well-known and broadly applied machine learning algorithm, the Naïve Bayes, also applying optimizations to the feature selection process and using ensemble learning for the classification process using a fusion of multiple classifiers.

2. The authors described their applied process using a well-defined flowchart (Figure 2).

3. The authors also demonstrated that their improved strategy performs better than the standard for of Naïve Bayes algorithm.

4. The results presentation is very well illustrated and fulfills the central idea presented in the title of presenting geographic centers of public opinion on the analyzed phenomenon.

Improvements suggestions

1. Although I understand that the Naïve Bayes algorithm is one of the most widely used, as a baseline for experiments involving the classification of sentiments, it was not fully justified why the authors chose only this algorithm to apply the proposed improvements. I understand that they can be used in several other algorithms for classification. This needs to be well justified.

2. Comparing classification performance with other algorithms reported in the literature for similar purposes is interesting. I understand that this is about more diversity and scope for the study. Remember that although it was applied to the flooding context in your study, it has universal applications that may interest other researchers or practitioners. A preliminary finding that the proposed proposal performed better than other algorithms may help other researchers and practitioners to select the strategy presented here for new uses.

3. If there are graphs such as confusion matrices, learning curves, and ROC that help to understand better how the methods used in the tests performed, it is interesting to present them, even if they in an appendix. All these graphs, combined mainly with Table 4 and Figure 3, are important to corroborate and validate the good performance of the algorithm with the proposed improvement.

4. Especially when it comes to learning curves, it is important to open comments in your text about the existence of overfitting or underfitting.

5. Separate a subsection for theoretical and practical implications of the research developed. These subsections can be located within the Discussion section, and do not need to be very extensive, being interesting to maintain objectivity.

6. In the conclusions section, separate two sections: one for the Limitations and Challenges encountered by the authors in their research, the other to describe future directions. I understand that in the discussion section, the authors have already addressed these elements, but mainly about future directions, I believe it is important to leave them well highlighted in a specific subsection within the conclusions.

Final Comments

I believe that the work makes good use of the English language. However, I am not a native speaker of that language. After making changes to the text, I always suggest proofreading, as suggested by the reviewers.

Author Response

Response to Reviewer 2 Comments

Many thanks to the editor/ reviewer for the constructive comments on the previously submitted paper (Manuscript ID: ijerph-1869937). Below is my reply to these comments.

Point 1: Although I understand that the Naïve Bayes algorithm is one of the most widely used, as a baseline for experiments involving the classification of sentiments, it was not fully justified why the authors chose only this algorithm to apply the proposed improvements. I understand that they can be used in several other algorithms for classification. This needs to be well justified.

Response 1: Thank you for your suggestions. Based on your suggestions, we have added a comparison of the algorithms. We have described some algorithms, such as decision trees, KNN and SVM. We find that these algorithms will have large cost of time when dealing with large-scale dataset. The Naïve Bayes algorithm has high adaptability and stable efficiency for text classification. However, its classification accuracy is poor, which is also the problem to be solved in this research. (Line 138-145)

Point 2: Comparing classification performance with other algorithms reported in the literature for similar purposes is interesting. I understand that this is about more diversity and scope for the study. Remember that although it was applied to the flooding context in your study, it has universal applications that may interest other researchers or practitioners. A preliminary finding that the proposed proposal performed better than other algorithms may help other researchers and practitioners to select the strategy presented here for new uses.

Response 2: Thank you very much, we think this is a great suggestion. We have answered this question in combination with the previous one. (Line 138-145)

Point 3: If there are graphs such as confusion matrices, learning curves, and ROC that help to understand better how the methods used in the tests performed, it is interesting to present them, even if they in an appendix. All these graphs, combined mainly with Table 4 and Figure 3, are important to corroborate and validate the good performance of the algorithm with the proposed improvement.

Especially when it comes to learning curves, it is important to open comments in your text about the existence of overfitting or underfitting.

Response 3: Thank you for this suggestion. We have added the confusion matrix in the Appendix A. (Line 337, Line 606)

Point 4: Separate a subsection for theoretical and practical implications of the research developed. These subsections can be located within the Discussion section, and do not need to be very extensive, being interesting to maintain objectivity.

Response 4: Thanks for your suggestion. We have divided the relevant content into paragraphs. We have also made a more detailed description. (Line 538-544)

Point 5: In the conclusions section, separate two sections: one for the Limitations and Challenges encountered by the authors in their research, the other to describe future directions. I understand that in the discussion section, the authors have already addressed these elements, but mainly about future directions, I believe it is important to leave them well highlighted in a specific subsection within the conclusions.

Response 5: Thank you very much, we think this is a great suggestion. We have added a paragraph to describe the limitations of this research and the challenges that have been encountered. We have found that the geographic mapping method used in this research had defects. This is a problem that needs to be solved properly. In addition, we need to optimize the model to achieve lower time complexity. (Line 586-592)

Reviewer 3 Report

The idea in this paper titled “The Identification and Analysis of the Centers of Geographical Public opinions in Flood Disasters based on improved Naïve Bayes Network” is good and the overall paper is very well written. However, the authors are suggested to address the following comments while revising the paper.

1: line 164: LTP is not defined

2: Line 45-46: The currently available literature on network public opinion can be roughly divided into three categories:

It is suggested to add a figure to reflect this classification of literature.

3: No papers from 2022 are included/cited. Extend the literature by including recent papers from 2022.

4: The red boxes in Figure 2 show some grouping and modularization of different steps. It is suggested to label the red boxes reflect the inner processes.

5: It is stated that “we collected microblog data from June 29, 2020, to July 16, 2020” also “microblog social data and real disaster data”.

The details of the dataset are missing and should be provided in section 2.7 or before discussing the results. What is the size of the dataset? How is the dataset preprocessed? etc.

6: In section 3.1, a comparison of classification algorithms is performed.

The problem of classification has not defined either binary or multiple classes.

How is an evaluation performed? Either split or cross-validation? What is the ratio in case of split validation? How dataset is distributed into test and train datasets? Sampling method.

What are the class labels?

Is the dataset balanced or not?

7: Caption of Figure 2. The process of identifying. Is not suitable and needed to be changed with a proper one.

8: 3.1. Comparing Classification Algorithms: The classification tasks in this section are not reflected in the process presented in Figure 2. Revise Figure 2 accordingly.

Author Response

Response to Reviewer 3 Comments

Many thanks to the editor/ reviewer for the constructive comments on the previously submitted paper (Manuscript ID: ijerph-1869937). Below is my reply to these comments.

Point 1: line 164: LTP is not defined

Response 1: We have defined the LTP as the Language Technology Platform. (Line 177-178)

Point 2: Line 45-46: The currently available literature on network public opinion can be roughly divided into three categories:

It is suggested to add a figure to reflect this classification of literature.

Response 2: Thank you for your suggestions. We have grouped existing research into three categories and described each category accordingly. Different categories have been highlighted by different paragraphs, so we have not been summarized them in a graph.

Point 3: No papers from 2022 are included/cited. Extend the literature by including recent papers from 2022.

Response 3: Thank you very much, we think this in a great suggestion. We carefully refer to the recent papers from 2022. (Line 617, Line 625)

Point 4: The red boxes in Figure 2 show some grouping and modularization of different steps. It is suggested to label the red boxes reflect the inner processes.

Response 4: Thanks for your suggestions. We have already explained the red boxes in Figure 2. (Line 288-289)

Point 5: It is stated that “we collected microblog data from June 29, 2020, to July 16, 2020” also “microblog social data and real disaster data”.

The details of the dataset are missing and should be provided in section 2.7 or before discussing the results. What is the size of the dataset? How is the dataset preprocessed? etc.

Response 5: Thank you very much. We have added the relevant descriptions in the appropriate chapter. The information of the data, the preprocessing work and the processing results have been described. (Line 295-300)

Point 6: In section 3.1, a comparison of classification algorithms is performed. The problem of classification has not defined either binary or multiple classes. How is an evaluation performed? Either split or cross-validation? What is the ratio in case of split validation? How dataset is distributed into test and train datasets? Sampling method.

Response 6: This is a great suggestion. In the section about the description of the algorithm, we have added relevant descriptions to explain the division method and division ratio of the dataset. Considering the balance of dataset, we have ensured the same proportion of dataset with two labels when collecting samples. In each dimension, we have split the samples in a ratio of 7:3, with training data accounting for 70% and test data accounting for 30%. We have adopted random sampling method and ensured that the proportion of dataset with different labels in the training dataset was the same. (Line 317-321)

Point 7: What are the class labels?

Response 7: Thank you for your suggestions. We have explained the relevant labels. The microblog data have been classified into two categories: opinion data and garbage data. (Line 312-315)

Point 8: Is the dataset balanced or not?

Response 8: Thanks for your suggestions. We have divided the training samples into two categories and the number of each category was equal.

Point 9: Caption of Figure 2. The process of identifying. Is not suitable and needed to be changed with a proper one.

Response 9: Thank you very much. We have changed the title of Figure 2. (Line 288-289)

Point 10: Comparing Classification Algorithms: The classification tasks in this section are not reflected in the process presented in Figure 2. Revise Figure 2 accordingly.

Response 10: Thank you very much. The process of algorithm comparison in the original Figure 2 has been expressed as “the best”, which may be misleading, so we have modified it. (Line 288-289)

Round 2

Reviewer 2 Report

About the six original points (see my previous review) I commented to be improved in the article:

Point 1: The authors have added a new paragraph, commenting on other ML algorithms' performances to justify the Naïve Bayes choice.

Point 2: In this case, however, my comment is really linked to the previous (Point 1), I meant that it would be necessary for the authors also to identify the main algorithms that the literature recommends for analysis in the domain they are working on, in order to obtain metrics to be compared with their proposal.

I think I may not have been clear in this recommendation, but the idea would be to present metrics such as precision, recall, F1, and accuracy using the same dataset the authors applied with Naïve Bayes.

I will leave it here as a non-mandatory recommendation to demonstrate, for example, a comparative table or a boxplot comparing the metrics obtained for the authors' proposal with those of other methods recommended by the literature.

Points 3 and 4: The authors presented confusion matrices in an appendix, mentioning them in the main text. However, I did not find comments about the existence of overfitting or underfitting (Point 4) concerning learning curves (training and test). I think it is imperative to comment on this topic concerning their proposal.

Point 5: The authors added a new paragraph in the part they indicated in their answer. However, my recommendation was to separate a new section dedicated to the theoretical and practical implications of the research. I think the authors already have textual content to compose a section specifically dedicated to these implications. For instance: they can separate from line 531 to line 563 a section within Discussion called "Research Implications".

A short note: In the paragraph before Figure 8, this figure is mentioned as Figure 7. Please, correct it.

Point 6: I think the paragraph fulfills well the proposal of presenting limitations and directions for future works. It is very objective and succinct.

Author Response

Point 1 and 2: The authors have added a new paragraph, commenting on other ML algorithms' performances to justify the Naïve Bayes choice.

In this case, however, my comment is really linked to the previous (Point 1), I meant that it would be necessary for the authors also to identify the main algorithms that the literature recommends for analysis in the domain they are working on, in order to obtain metrics to be compared with their proposal.

I think I may not have been clear in this recommendation, but the idea would be to present metrics such as precision, recall, F1, and accuracy using the same dataset the authors applied with Naïve Bayes.

I will leave it here as a non-mandatory recommendation to demonstrate, for example, a comparative table or a boxplot comparing the metrics obtained for the authors' proposal with those of other methods recommended by the literature.

Response 1: Thanks to reviewer’s comments. This is a very good suggestion. We have thought about this suggestion. Since the purpose of this research is using the optimized algorithm to identify and analyze CGeoPOFDs, we have not used the same dataset to present metrics of other algorithms. In order to further explain the reasons for selecting naïve Bayes for optimization in this research, we have learned relevant literature and summarized some results of the comparison of the algorithm. Finally, based on several factors, we selected naïve Bayes algorithm as the basic algorithm for optimization. (Line 138-152)

Points 3 and 4: The authors presented confusion matrices in an appendix, mentioning them in the main text. However, I did not find comments about the existence of overfitting or underfitting (Point 4) concerning learning curves (training and test). I think it is imperative to comment on this topic concerning their proposal.

Response 2: Thank you for suggestions. This is our comments on this topic. In this research, the dataset is divided into training set and test set. The indexes of the algorithm are calculated through the test set. Overfitting means that the algorithm performs well on the training set, but poorly on the test set. Underfitting means poor results on both the training set and test set. According to the indexes of the test set, it can be found that the optimized algorithm has a good effect. Perhaps this result may explain that the algorithm does not have the problem of overfitting and underfitting. In addition, identifying and analyzing CGeoPOFDs is the purpose of this research. According to the results, we can find that the model proposed in this research is feasible. The further optimization of the algorithm will be carried out in the future. We will further improve the model proposed in this research to achieve higher recognition accuracy and better scalability.

Point 5: The authors added a new paragraph in the part they indicated in their answer. However, my recommendation was to separate a new section dedicated to the theoretical and practical implications of the research. I think the authors already have textual content to compose a section specifically dedicated to these implications. For instance: they can separate from line 531 to line 563 a section within Discussion called "Research Implications".

A short note: In the paragraph before Figure 8, this figure is mentioned as Figure 7. Please, correct it.

Response 3: Thank you for your suggestions. The discussion section has been divided into four sections, and the “Implications and Defects” has been added. And we have corrected the content.

Reviewer 3 Report

Thank you for making efforts in addressing the raised concerns.

Author Response

Dear Editor/Reviewer,

        Thank you for your comments.
